

# Characterization of bottom sediment resuspension events observed in a micro-tidal bay

Manel Grifoll[1,2], Pablo Cerralbo[1,2], Jorge Guillén[3], Manuel Espino[1,2], Lars Boye Hansen[4], Agustín Sánchez-Arcilla[1,2]

[1]Laboratori d'Enginyeria Marítima, Universitat Politècnica de Catalunya (UPC-BarcelonaTech), Barcelona, 08034, Spain
[2]Centre Internacional d'Investigació dels Recursos Costaners (CIIRC), Barcelona, 08034, Spain.
[3]Institu de Ciències del Mar (ICM-CSIC), Barcelona, 08003, Spain.
[4]DHI-Gras, Horsholm, DK-2970, Denmark.

*Correspondence to*: Manel Grifoll (manel.grifoll@upc.edu)

**Abstract.** In this contribution we investigate the origin of the variability in near-bottom turbidity observations in the Alfacs Bay (NW Mediterranean Sea). This bay is characterized by a micro-tidal environment and a relevant seiching activity which may lead to flow velocities of more than 50 cm•s-1. A set of current meters and optical sensors mounted near the sea bottom were used to acquire synchronous hydrodynamic and optical information of the water column. The time-series observations showed an evident relation between seiche activity and sediment resuspension events. The observations of turbidity peaks are consistent with the node/anti-node location for the fundamental and first resonance periods of the bay. The implementation of a coupled wave-current numerical model shows a strong spatial variability of the potential resuspension locations. Strong wind events are also a mechanism responsible of the resuspension of fine sediment within the bay. This is confirmed using retrieval of suspended sediment concentration from Sentinel-2 data. We suggest that the sequence of resuspension events plays a relevant role in SSC, in such a way that previous sediment resuspension events may influence the increase of suspended sediment in subsequents events. The suspended sediment events likely affect the ecological status of the Bay and the sedimentary process at long-term period.

## 1 Introduction

Suspended sediment in the water column and subsequent deposition plays a critical role in coastal environment and management. High levels of suspended sediment concentration in the water column has relevant implications in aquatic ecosystem and natural habitat (Ellis et al., 2002) in particular during large exposure periods ((Newcombe and Macdonald, 1991). Also, sediment supplied from rivers convey load of organic matter, pollutants and heavy metal that may be deposited in the vicinity sea bottom or transported offshore (Palanques et al., 2017). The sediment dynamics is relevant in coastal bays and estuaries due to the large amount of sediment delivered by the freshwater and the potential fine sediment trapping zones. In addition, sediment resuspension can results in a large contribution to the total nutrient load (Sondergaard et al., 1992) and



prevent the sunlight penetration (Mehta, 1989). Besides, the analysis and prevention of fine sedimentation within basins and channel access is object of investigation in port engineering context in order to examine the siltation process (e.g. (Ghosh et al., 2001; van Maren et al., 2015)). Finally, harmful species, such as dinoflagellate cysts, may be related to significant local resuspension through the mixing of the upper layers, resulting to more homogenous cyst profiles in the sediment (Giannakourou et al., 2005).

In coastal area, the transport of fate of sediment is related with the hydrodynamic pattern. On large time scales advection processes redistribute and determine the final depositional pattern in function of the sedimentological and water current variables (Bever et al., 2009; Ogston et al., 2000). Hydrodynamics processes such as wind-waves (Carlin et al., 2016; Grifoll et al., 2013), tides (Fan et al., 2004; Garel et al., 2009), winds (Hofmann et al., 2011; Sherwood et al., 1994), surface seiches (Jordi et al., 2008) or internal-seiches (Shteinman et al., 1997) promote the resuspension, advection and settling of fine sediment conditioned by the continental sediment sources. Subsequent resuspension effects due to natural causes also contributes at the reworking and final deposition of the sediment load (Grifoll et al., 2014a; Guillén et al., 2006). In this sense, anthropogenic activities such as, fishing trawling, ship propellers and waves generated by vessels may bring additional energy at the water system influencing the resuspension, transport and final sediment deposition in shallow waters (e.g. (Garel et al., 2009; Hofmann et al., 2011).

Alfacs Bay (NW Mediterranean Sea; Southern part of the Ebro Delta) is a micro-tidal estuary. It has been investigated extensively in the past in terms of hydrodynamics response (Cerralbo et al., 2015a, 2016; Llebot et al., 2014; Solé et al., 2009), tidal wave propagation (Cerralbo et al., 2014), biochemical processes (Llebot et al., 2010, 2011) an optical water properties (Ramírez-Pérez et al., 2017). The estuary receives freshwater discharge from the rice fields of the Ebro river. Thus, Alfacs Bay is an intensively exploited area with tourism, fishing and aquaculture activities being an ecosystem of relevant economic importance in the region. Several episodes of algal blooms (linked with the increase of nutrients and perhaps triggered by resuspension mechanisms) and presence of harmful bacterium in bivalve with negative effects on aquaculture have been reported (Loureiro et al., 2009; Roque et al., 2009).

With the purpose to improve the knowledge in fine sediment dynamics in coastal bays, the goal of this investigation is to provide a physical interpretation of the sediment resuspension events observed within a micro-tidal bay (Alfacs Bay; NW Mediterranean Sea). Using sea-level, water currents and wind measurements we investigate the driven mechanisms that resuspend fine bottom sediment within the bay. Then, the spatial and temporal interpretation of the resuspension mechanisms linked with the hydrodynamics is analyzed through the implementation of a wave-current coupled numerical model. The contribution aims to provide explanation of resuspension mechanisms; the knowledge of these mechanism may have an evident benefit for human activities management mentioned previously (e.g. harmful species resuspension or algal blooms with negative effects on aquaculture activities).



The water circulation in Alfacs Bay has been widely analyzed in previous contributions using observational data set and numerical results (Camp and Delgado, 1987; Cerralbo et al., 2014, 2015a; Llebot et al., 2014). However, fine sediment dynamics and its resuspension mechanisms has not been examined yet. Synchronous optical measurements, jointly with

velocity and sea-level measurements, has entailed a good chance to advance in the interpretation of resuspension mechanisms in Alfacs Bay. This area is an example of micro-tidal estuary, thus being the wind or wind-waves candidates mechanisms of fine sediment dispersal.

## 2 Methods

### 2.1 Study Area

Alfacs Bay, located at south of the Ebro delta, is formed by the prograding shouthern spit. The semi-enclosed bay is about 16 km long and 4 km width. The average depth is 4 m and the maximum depth is about 6.5 in the middle of the Bay (Figure 1). The connection with the open sea is 2.5 km, with a central channel of 6.5m and shallow edges of around 1-2 m on both sides. The bay is surrounded by rice fields to the north, which spill around 10 $m^3 \cdot s^{-1}$ of freshwater loaded with nutrients during 9-10 months per year (April-December) distributed in several channels, and a sand beach closing it on the east side. The seabed in

the central part of the bay is composed by very fine sediment (typically 65-65% silt, 30-35% clay and around 5% sand) increasing the sandy content towards the edges of the bay (Guillen and Palanques, 1997; Satta et al., 2013). The bottom sediment of Alfacs is composed by mud and sandy mud, with significant content of clay (Palacín et al., 1991). They found that the muddy sediment extended by the central part of the bays and the content of sand increased near to both spits that separate bays from open sea and also in the southern shallow edge.

The bay has been defined as a salt-wedge estuary (Camp and Delgado, 1987) with almost stable stratification all year. The highest tidal range during spring tides is around 0.2 m, and the hydrodynamic fluctuations are controlled by the wind modulated by the seiche activity in a short periods (Cerralbo et al., 2015a). The water circulation in the low-frequency band is dominated by both winds and salinity gradients due to freshwater discharge (Solé et al., 2009). The most intense regional winds in the

area are from the north and northwest, establishing a wind jet due to the orographic effects in the Ebro River valley (Grifoll et al., 2015, 2016). This offshore wind is characterized by noticeable spatial variability due to the surrounding topography (Cerralbo et al., 2015b). The water column within the bay used to be stratified due to the freshwater discharge, but well-mixed conditions are common during winter as a consequence of the hydrodynamic response to strong wind forcing (Llebot et al., 2014) and occasionally to seiches (Cerralbo et al., 2015a). During summer, the contribution of the temperature at the

stratification may be also substantial (Cerralbo et al., 2015a).



### 2.2 Measurements campaigns

The bulk of the observational data correspond at two months field campaign from July to mid-September 2013 that correspond to summer conditions. The data set consisted of water currents from two 2MHz Acoustic Doppler Current meter Profiler (ADCPs) moored in the mouth (A1) and inner bay (A2) (Fig.1) configured to record 10 min averaged data from 10 registers per minute and with 25 cm vertical cells. Both devices were equipped with Optical Backscatter Sensor (Campbell Scientific OBS-3), bottom pressure meter and a temperature sensor, and they were mounted on the sea bottom at 6.5 m depth. OBS signal is transformed to Nephelometric Turbidity Units (NTU) using device calibration. Besides, the study area used to present a linear relation between optical signal and suspended sediment concentration (Guillén et al., 2000). The distance of the ADCPs and OBS sensor were 0.25 m above the sea bed. The ADCP has a 20 cm of blanking zone. Additional sea level data were obtained through a sea level gauge mounted in Sant Carles de la Ràpita harbor (Fig.1) and bottom pressure systems from the ADCPs. Atmospheric data (wind, atmospheric pressure, solar radiation and humidity) were obtained from a fixed land station located in Sant Carles de la Ràpita (M-Sc).

### 2.3 Current and wave model implementation

We use the coupled version of SWAN-ROMS models included in the COAWST system in order to simulate the hydrodynamics within the bay. The COAWST system (Warner et al., 2010) consists of several state-of-the-art numerical models that include ROMS (Regional Ocean Modeling System) for ocean and coastal circulation and SWAN (Simulating Waves Nearshore) for surface wind-wave simulation. SWAN is a third-generation numerical wave model that computes random, short-crested waves in coastal regions with shallow water and ambient currents (Booij et al., 1999). It is based on the wave action balance with sources and sinks and incorporates the state-of-the-art formulations of the processes of wave generation, dissipation and wave-wave interactions. ROMS is a three-dimensional circulation model which solves the primitive variables on a sigma-level in the vertical and horizontal curvilinear grid. Numerical aspects of ROMS are described in detail in (Shchepetkin and McWilliams, 2005). In COAWST system, the wave model provides hydrodynamic parameters (i.e., significant wave height, average wave periods, wave propagation direction, near-bottom orbital velocity and wave energy dissipation rate) to the water circulation model. The ocean model provides water depth, sea surface elevation, and current velocity to the wave model. The variables exchange is made "on-line" during the simulation processes, via Model Coupling Toolkit (Jacob et al., 2005), where a multi-processes MPI protocol is used to distribute the computations among several nodes. The COAWST also include different formulations to parametrize the wave-current bottom boundary layer and the wave effect on currents (Kumar et al., 2012; Warner et al., 2008).

The implementation of the COAWST system in Alfacs Bay consisted of a regular grid of 186 x 101 points with a spatial resolution of 100 m (in both x and y) and 12 sigma levels in the vertical. Details of the implementation and the skill assessment of the ROMS model in Alfacs bay is provided in (Cerralbo et al., 2015a). The same regular grid is used by the SWAN model.



A two-year water circulation simulation (2012-2013) was performed in order to obtain realistic three-dimensional temperature and salinity fields. The interval time between change of variables of ROMS and SWAN was established in 3600 s. For both simulations, open boundary was forced with depth-averaged velocities and sea level measured at A1 (10' data). The freshwater inputs are distributed on 8 points simulating the main rice channels with a total flow of 10m3s-1 (see (Cerralbo et al., 2015a)).

The bottom boundary layer was parameterized using the combined wave-current (Styles and Glenn, 2000) adopted in ROMS and SWAN coupling in (Warner et al., 2008). The input parameters for the model are the velocitiy components near the bottom and wave characteristics near the bottom (wave period, wave direction and the wave orbital direction). For each computational step, an initial assessment of bed roughness length is estimated in function of the grain size, ripples and sediment transport. Then, the pure current ($\tau_c$) and pure wave ($\tau_w$) bottom stress are computed as:

$$\tau_c = \frac{(u^2+v^2)\kappa^2}{\ln^2(z/z_0)}$$ (1)

$$\tau_w = 0.5f_w u_b^2$$ (2)

where z is the vertical coordinate, $\kappa$ is the von Karman's constant, and $f_w$ is the Madsen wave-friction factor. Then, the maximum bottom stress under wave-current conditions is computed as (Soulsby, 1997):

$$\tau_{wc} = \tau_b \left(1 + 1.2\left(\frac{\tau_w}{\tau_w+\tau_c}\right)^{1.5}\right)$$ (3)

The Eddy profiles are scaled in the wave-boundary layer in outer boundary layer following and iterative processes. The wave effects on currents are considered using vortex-force formalism, which is included in COAWST. This approach allows to consider the effect of the gravity waves on the mean flow and was tested in different experimental and real configurations by (Kumar et al., 2012).

## 3 Results

### 3.1 Observations

In order to investigate the suspended sediments events within Alfacs Bay we use a sub-set of the total observations recorded in A2: from 2nd August to 8th August 2013. This is because the sub-set data selected include the main hydrodynamic conditions susceptible to increase the near-bottom turbidity. Figure 2 show the time-series recorded in A2 in terms of NTU from the OBS, sea level height measured (additionally sea-level height measured in A1 is also shown), bottom current intensity in m•s-1 in A1 and wind intensity and direction measured in M-Sc.



The wind characterization (Figure 2.a and 2.b) include two of the most typical situations in the region: sea breeze and the NW winds (Cerralbo et al., 2015a). The sea breeze is associated to an increase of wind intensity during the central hours of the day

(approximately from 11:00 hr to 18:00 hr with a wind direction within the range 30º to 180º approximately). From a daily point of view, this seems evident during the 1$^{st}$ to 6$^{th}$ of October. A different pattern is observed during the wind intensity peak of 7$^{th}$-8$^{th}$ of August where 330º wind direction were measured. This corresponds to an offshore wind typical from the region (NW winds called "Mestral").

The period of analysis, also include a seiche event during the 3$^{hd}$ of August. This seiche event was previously characterized

hydro-dynamically in Cerralbo et al., (2015a) revealing a characteristic oscillation of 1 hour period in sea-level and currents. This oscillation is characterized by a node (approximately located at A2) where the velocities are maximum, and an anti-node (approximately located in A1) where the amplitude is maximum (see sea-level height in A1 in comparison to A2 in Figure 2.c). The homogeneous vertical profile in velocities measured in A2 is shown in Figure 3, where the along-shore direction reveal velocities peaks of the order of 0.5 m•s-1 in the water column. The near-bottom water current intensities in A2 (Figure

2.d) show fluctuations with peaks over 0.1 m•s-1 excepting the mentioned seiche event where peaks arising 0.4 m•s-1.

The near-bottom turbidity shows a fluctuating behavior ranging values from almost zero to higher than 10 NTU (Figure 2.d). In this sense, three differentiated events with high turbidity are observed. These events are E1 (covering from 3$^{th}$ of August to the first hours of 5$^{th}$ of August), E2 (6$^{th}$ of August) and E3 (between 7$^{th}$ and 8$^{th}$ of August). The maximum turbidity is measured during the E1 (maximum turbidity 41.1 NTU). This event lasts for a longer time in comparison to E2 (with a maximum

turbidity 4.6 NTU) and E3 (maximum turbidity 12.1 NTU).

### 3.2 Skill assessment near the sea bottom

The performance of the water circulation model used in this contribution was examined in terms of sea-level, water currents and temperature/salinity evolution in previous works (Cerralbo et al., 2014). However, in this work we pay attention to the near-bottom velocities because they have a relevant role in the sediment transport dynamics and as a mechanism of

resuspension. Thus, the skill assessment of the near-bottom velocities in A1 and A2 is analyzed using Taylor diagram (Taylor, 2001). This diagram characterizes the similarity between numerical model and observations using their correlation, the root-mean-square difference (RMSD) and the amplitude of their variations (represented by their standard deviations). The model skill improves as the points get closer to the observation reference point in the diagram. In general, the model results showed a good agreement with the observations in the prevalent along-shelf direction, with correlations larger than 0.5 and RMSD

below 1. In addition, the water current fluctuations are well represented in the model because the normalized standard deviation is closer to 1 in both measuring points.

### 3.3 Modelled bottom stress

The bottom stress is obtained from the coupled numerical model implemented in Alfacs Bay. The Figures 5 and 6 show different snapshots in order to examine the bottom stress pattern for both components (i.e. wave and current-induced bottom





stresses). These snapshots corresponds to different episodes identified from the previous observational analysis. The plot scale
of the bottom stress is transformed in log10 for clarity. During the case E1 (3$^{hd}$ of August 2013; 10:00 hr) the combined bottom
stresses are mainly due to the current bottom stress (Figure 5). Maximum values of 0.15 Pa for the combined bottom stress are
obtained in the center of the bay and the mouth. This episode corresponds to a seiche event and the spatial variability of the
bottom stress is consistent with the spatial pattern of the node/antinode position. It means that the maximum combined bottom

stress (associated at maximum water currents) corresponds to the node position (minimum sea-level amplitude). In opposite,
the minimum bottom stress to the antinode position (maximum sea-level amplitude). The position A2 is located near to the
node, where the water currents are maximum during the seiche event (0.08 Pa for combined bottom stress). It is worth to
mention the node/antinode pattern of the current-induced bottom stress, which presumably would indicate a large spatial
variability on the resuspension process within the Bay.

After the seiche activity (second stage of E1 where the wind intensity increase due to the sea-breeze), the current-induced
bottom stress decreases significantly in particular in the center of the Bay (Figure 5). The bottom stress distribution shows how
the maximum values are obtained near the shoreline (2.2 Pa) due to the contribution of the wave-induced bottom stress. In A2,
the combined bottom stress is equal to 0.03 Pa (value presumably far to induce resuspension). For this event, the wave field
during the sea-breeze is shown in Figure 7. This figure shows how the maximum significant wave height (equal to 0.3 m)

occurs near the northern and southern shallow edge consistent with the maximum wave-induced bottom stress.
The bottom-stress pattern during the episode E2 (Figure 6) is similar to the second stage of the episode E1. Both wave and
current bottom stress tends to be small in A2 in comparison to the seiche event. Only substantial bottom stress are observed in
the shallow edges of the Bay due to the wave action originated by the sea-breeze.
During the episode E3 (NW wind, Figure 6), the combined bottom stress is dominated by both wave and current action. The

southern part of the bay shows the maximum wave induced bottom stress consistent with the wave climate (Figure 7). Also,
the current induced bottom stress presents non negligible values within the bay. Focusing in A2, both mechanism contribute
in similar manner (wave and current bottom stress is 0.09 and 0.06 Pa respectively) in the combined bottom stress.

## 4 Discussion

The synchronous time-series of the metro-oceanographic variables and turbidity shown in Figure 2, jointly with the bottom

stress modelled provides a good opportunity to characterize the turbidity peaks measured in A2. During the first stage of the
episode E1, the bottom current intensity responds at the node-antinode pattern with velocities that raise 0.4 m•s-1 in A2.
Apparently, the high turbidity event in E1 is correlated with an increase of the bottom velocity at A2 and it can interpreted as
a bottom sediment resuspension event (Figure 2) . Even that an increase of wind intensity occurs (peaks that raise 8 m•s-1),
the oscillating pattern of the current (see Fig.3), strongly polarized following the along-shore direction with 1-hr period, suggest

an increase of turbidity due to the seiche instead of wind driven current. The bottom stress modeled during E1 (Figure 5) also
suggested that the seiche is the main mechanism for turbidity increase in A2 during the first stage of event E1. In consequence,





the eventual seiche is a clear mechanism of fine sediment resuspension in Alfacs Bay. Resupension mechanism in water environments caused by seiches are suggested in observational investigations (Chung et al., 2009; Jordi et al., 2011; Niedda and Greppi, 2007). However, the numerical results of the current-induced bottom stress shown in Figure 5 suggest a high

spatial variability of the seiche-induced resuspension not examined in the mentioned contributions. It means observational results may differ significantly in function of the location of the node/anti-node and its consequent maximum and minimum velocities.

The turbidity still shows large values after the seiche was already dissipated and the bottom current decreased. During the second stage of the E1 event, a typical sea-breeze wind conditions were observed (gentle variation of wind direction from 30º

to 180º, with a noticeable increase of the wind intensity during 4th of August). However, increase of wind intensity does not show a correlation with the current bottom intensity measured. (Llebot et al., 2014) and (Cerralbo et al., 2015a) stated that water current profile due to winds observed in Alfacs Bay does not imply a barotropic shape in the water column. Several authors suggest near the bottom a differentiate behaviour than surface related to wind set-up phenomena. In consequence, the local resuspension due to wind-breeze seems unlikely. It seems more feasible that the high turbidity measured in A2 during

this event is associated to advection of fine sediment resuspended previously (likely during the first stage of episode E1, it means seiche) or resuspended fine sediment in the shallow edges of the bay by the sea-breeze activity with a subsequent transport towards the middle of the Bay. This mechanism would explain also the turbidity peak measured during the 5th of August at 00:00; after the fine sediment settling occurred within the Bay. The sediment advection within the Bay is difficult to confirm according to our data set, but Alfacs bathymetry shows a characteristic shallow edge near the coastline (water depths

below 2 m; see Figure 1). In these shallow edges the bottom stress arise 0.8 Pa, suggesting a potential resuspension towards the center of the basin. This shallow edge may be a source of fine sediment under energetic wind conditions in case of fine sediment availability. In consequence, the advection of resuspended sediment highlight the relevance of the water current patterns within the Bay for turbidity measurements.

The episode E2 is associated at sea-breeze mechanism. This event is qualitatively less important in terms of turbidity measured

in A2. The comparison of the sea-breeze event during 4th of August and 6th of August (both have similar wind and bottom current intensities but different turbidity values) seems to indicate the importance of the previous events and the role that plays the advection of fine sediment in similar terms that was explained previously. Similar to the second stage of E1, in central basin of the Bay, the bottom stress are small (below 0.02 Pa); so the local resuspension is unlikely. In consequence, the turbidity measured in A2 is probably due to advection processes of suspended sediment from the shallowest areas (combined bottom

stress more than 0.8 Pa) to the central basin.

Finally, episode E3 correspond to a strong NW wind event with intensities that raise 12 m•s-1. However, during this episode the bottom current intensity does not show significant values in comparison to calm periods. In opposite to sea breeze, the sea waves generated by the NW wind conditions may have a relevant role in the resuspension mechanisms due to an increase of the wave induced bottom stress. According to the bottom stress shown in Figure 6, the wave induced bottom stress prevails,

but the complexity of the mechanism where the advection may have a relevant role, difficult to sort the importance of both

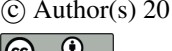



mechanism. Unhopefully, the set-up of the ADCP did not allowed to record the oscillatory pattern derived from the orbital velocities generated by waves.

E2 and E3 are examples of two mechanism that may produce local resuspension: wind-driven current and wind-wave resuspension due to an increase of the wave-induced bottom stress. In Alfacs Bay, these mechanisms are less clear in comparison to seiches because they are in function of wind intensity without a clear correlation between wind module and the turbidity observed. The resuspension of fine sediment due to wind and wind-waves in shallow environments have been reported in the literature (Bever et al., 2011; Grifoll et al., 2014b; Guillén et al., 2006; Hawley et al., 2014; López et al., 2017; Luettich et al., 1990; Martyanov and Ryabchenko, 2016; Ogston et al., 2000). Some of these works highlight the complexity of the sediment processes due to the temporal and spatial variability of the resuspension mechanisms and the presence of available material to be resuspended. This is the case of our observations, because similar wind conditions does not imply the same suspended sediment measurements. A good example is the sea-breeze wind events during 4th, 5th and 6th of August in which different turbidity values are observed. As we mention in the previous section, advective fluxes and the past event sequence may have a relevant role in the observed water turbidity. In this sense, many authors have reported an evident influence on advective fluxes correlated with suspended sediment concentration after an initial deposition of fine sediment (Bever et al., 2009; Grifoll et al., 2014b; Guillén et al., 2006; Harris et al., 2008; Ogston et al., 2000; Sherwood et al., 1994). This means that on longer time scales, advection of sediment by currents may redistribute sediment and determine final deposition patterns (Wright and Nittrouer, 1995). This may be the mechanism responsible of suspended sediment concentration observed under relative low hydrodynamic conditions. For instance, the fact that during the sea-breeze event of 2nd August does not appears sediment concentration in opposite to 4th (E1) of 5th of August may response at this mechanism where an energetic event (i.e. seiche) may mobilize sediment that after is resuspended easily in subsequent events. This lack of proportionality of the resuspension related to hydrodynamics is also found in extended data time-series where divergences are associated mainly at sediment availability in the bottom among other factors (e.g. in (López et al., 2017; Wiberg et al., 1994)). In the case of Alfacs Bay, more extended observations may clarify the relation between wind intensity, wind-waves and the amount of suspended sediment and fluxes taking into account the sequence of energetic events.

The sediment distribution in Alfacs Bay (high percentage of silt and clay in the central basin and sand prevalence in the southern, eastern and western shore) is consistent with the modeling results shown in this contribution, where larger bottom stresses were obtained in the lateral shallow edges due to the contribution of the wave induced bottom stress in shallow areas. However, as we mention previously, the deposition mechanism may be a complex process including an initial settling and a subsequent dispersal in a similar pattern to described in (Wright and Nittrouer, 1995). Further sediment transport simulations, including sediment classes and erosion and settling effects, would help to investigate the sediment settling dynamics and its final deposition. These processes must take into account the cohesive nature of the fine sediment, such as others phenomena such as the armoring and the bioturbation which may modify the physical properties of the sediment layers (Amoudry and Souza, 2011; van Ledden et al., 2004).



The bay geometry characteristics (for instance the relative narrow and shallow entrance) suggest the trapping effect of fine
sediment delivered by the freshwater outflow or the link between the open sea and the inner bay. The trapping effect of the
bay may entailed the presence of a thin surface layer of fine sediment easily involved in resuspension. This behavior is typical
from shallow and sheltered environments such as lagoons or lakes. According to (Luettich et al., 1990) or (Hofmann et al.,
2011), the regular resuspension events in sheltered and shallow water bodies prevent the sediment consolidation and the
formation of a cohesive sediment layer. This may be consistent with the turbidity values observed in the Alfacs bay under
relative weak conditions such as sea-breeze events, as opposite to be expected if the sediment was cohesive.

The image with unprecedent resolution obtained by the Sentinel-2 should allow to identify scenarios with resuspension linked
to hydrodynamic forcings. Figure 8 show the Total Suspended Matter (TSM in mgr•l-1) for the Alfacs Bay in two differentiate
scenarios: NW wind and Calm conditions. Without access to local calibration data, a generalized approach for TSM retrieval
has been applied. Through SNAP (v. 6.0.0) the Level 1C Sentinel-2 MSI data was converted to geophysical values (sediment
concentrations) using the most recent version of the water quality processor 'C2RCC' (v. 1.0). The C2RCC processor was run
using default values. Following processing in SNAP the data was post-processed (tiles merged and data noise corrected) and
the TSM maps created. NW wind conditions increase substantially the TSM in the southeastern shallow edges. This would be
a source of a subsequent advection of fine sediment towards the central Basin as it was stated in the previous paragraphs. In
opposite, calm conditions the values of TSM decrease significantly.

Also, the proximity of the Ebro river mouth (km at north) may increase the suspended sediment within the bay under particular
circumstance. River discharge is the main driver of the Ebro River plume, followed by wind and regional oceanic circulation
that tends to be southward (Fernández-Nóvoa et al., 2015; Mestres et al., 2003). Analysis of the turbid plume by remote sensing
products indicate that more than 70% of the plume extension was located south of the river mouth influenced by the regional
oceanic circulation (Fernández-Nóvoa et al., 2015). Others external sediment sources may be associated freshwater discharge
from channels, overwash in the bar, flash flood from small rivers or aeolian transport. The complete study of the suspended
sediment dynamics will provide objective information to address the problem of degrading water quality within the bay and
how to make use of natural mechanisms to limit undesired concentrations of nutrients or pollutants. This applies in particular
to harmful algae blooms prone to occur in the area under present and future conditions.

**5 Conclusions**

The observational set and the wave-current numerical results obtained for Alfacs Bay have permitted to investigate the
resuspension mechanisms of fine sediment. The results evidence a clear mechanism of resuspension induced by eventual seiche
events, which according to the bottom stress patterns may have a relevant spatial variability within the Bay. The wind and
wind-wave mechanisms also are responsible of fine sediment resuspension during energetic wind events. The relevance of the
sequence of such events is highlighted in the analysis linked with the effect of advective sediment fluxes within the bay (from
the lateral shallow edges to the deeper basin). In any case, the trapping effect of the bay may entail the presence of a thin



surface layer of fine sediment continuously involved in resuspension neglecting the expected cohesive effects. However, these points deserve further analysis with extended data sets and sediment transport modeling. The exchange of fine sediment within the Bay and the open sea seems also evident according to remote sensing images. As a region of high-anthropogenic pressure, this contribution may contribute to develop better integrated plans in the context of sustainable aquaculture activities and the
mitigation of the effects of climate change in the Ebro Delta.

**Acknowledgments**

The authors are grateful for the collaboration IRTA staff for the participation in the field campaigns carried out within the framework of the monitoring program of water quality at the shellfish growing areas in Catalonia. Thanks to the data provided by Puertos del Estado and AEMET. This work received funding from the EU H2020 program under grant agreement no.
730030 (CEASELESS project). We also want to thank to Secretaria d'Universitats i Recerca del Dpt. d'Economia i Coneixement de la Generalitat de Catalunya (Ref 2014SGR1253) who support our research group. The paper contains modified Copernicus Sentinel data [2017/2018].

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





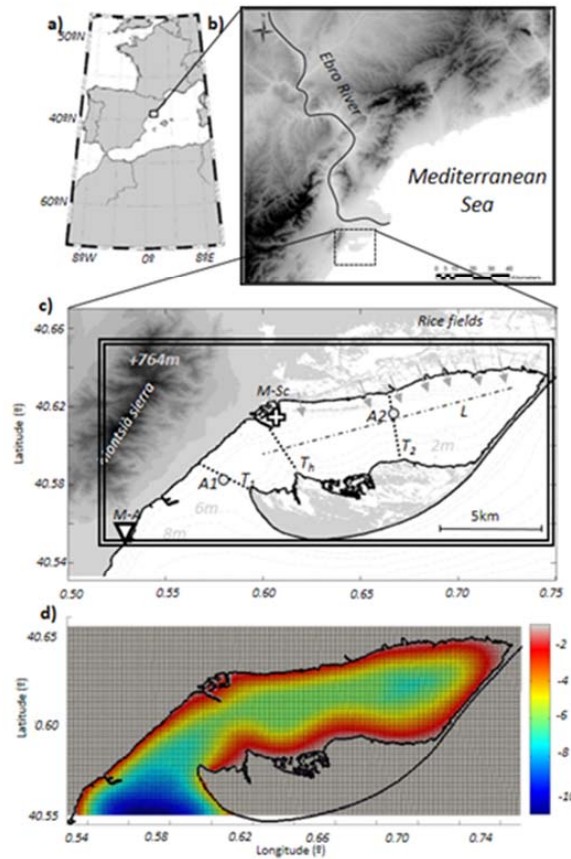


**Figure 1: a: Regional location of Ebro River Delta b: Alfacs Bay in Ebro River Delta. c: map of Alfacs Bay. Triangle shows the meteorological station (M-Sc). White cross for Sant Carles de la Ràpita tide gauge. Gray circles shows the ADCP and OBS mooring locations (A1 and A2). Gray arrows on the northern coast shows the freshwater drainage points considered in the simulation. Double line square indicate the domain for the hydrodynamic numerical model, which is shown in detail in image d (colorbar indicates**
**depth in meters).**

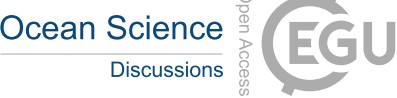



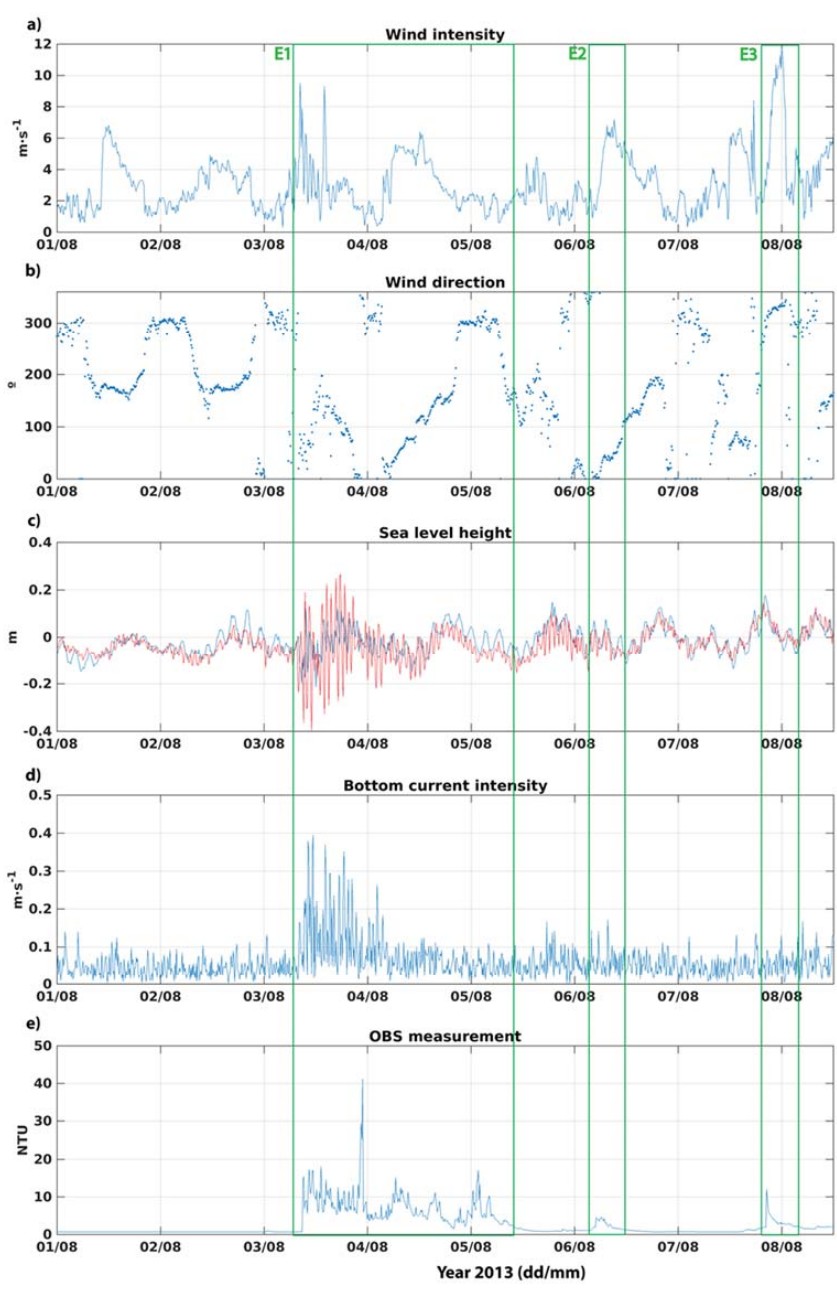





**Figure 2: Time-series of the variables measured during the field campaign. (a) wind intensity measured in M-Sc. (b) wind direction measured in M-Sc. (c) Sea-level height measured in A2 (blue) and A1 (red). (d) near-bottom current velocity measured in A1. (e) NTU measured in OBS mounted at A2 station. Vertical bars show the episodes considered in the analysis.**

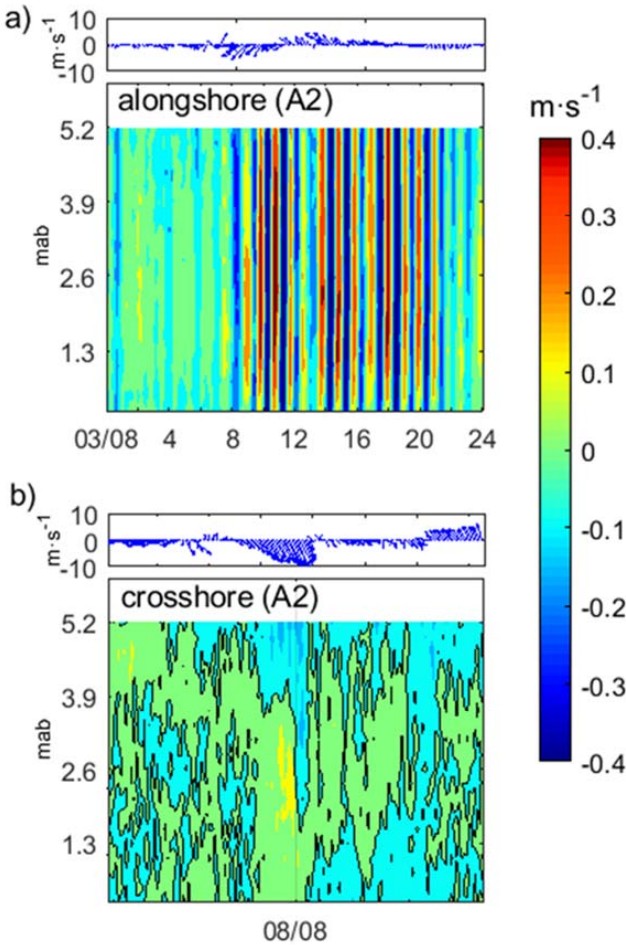


**Figure 3: Each panel shows on the top the wind measured at M-Sc (in m·s⁻¹), and on the bottom the vertical profiles velocities measured at A2 (in m·s⁻¹). In each panel, different events are showed: 3/8/2013 for alongshore velocities (panel a) and 8/8/2013 for crosshore (panel b). Black lines show 0 velocity isolines.**



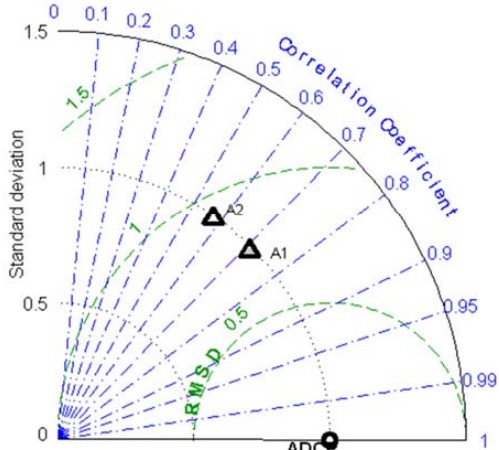

**Figure 4: Taylor diagram comparing the error metrics between the observations and model results for the near-bottom currents. A1 and A2 corresponds to the ADCP locations shown in Figure 1.**




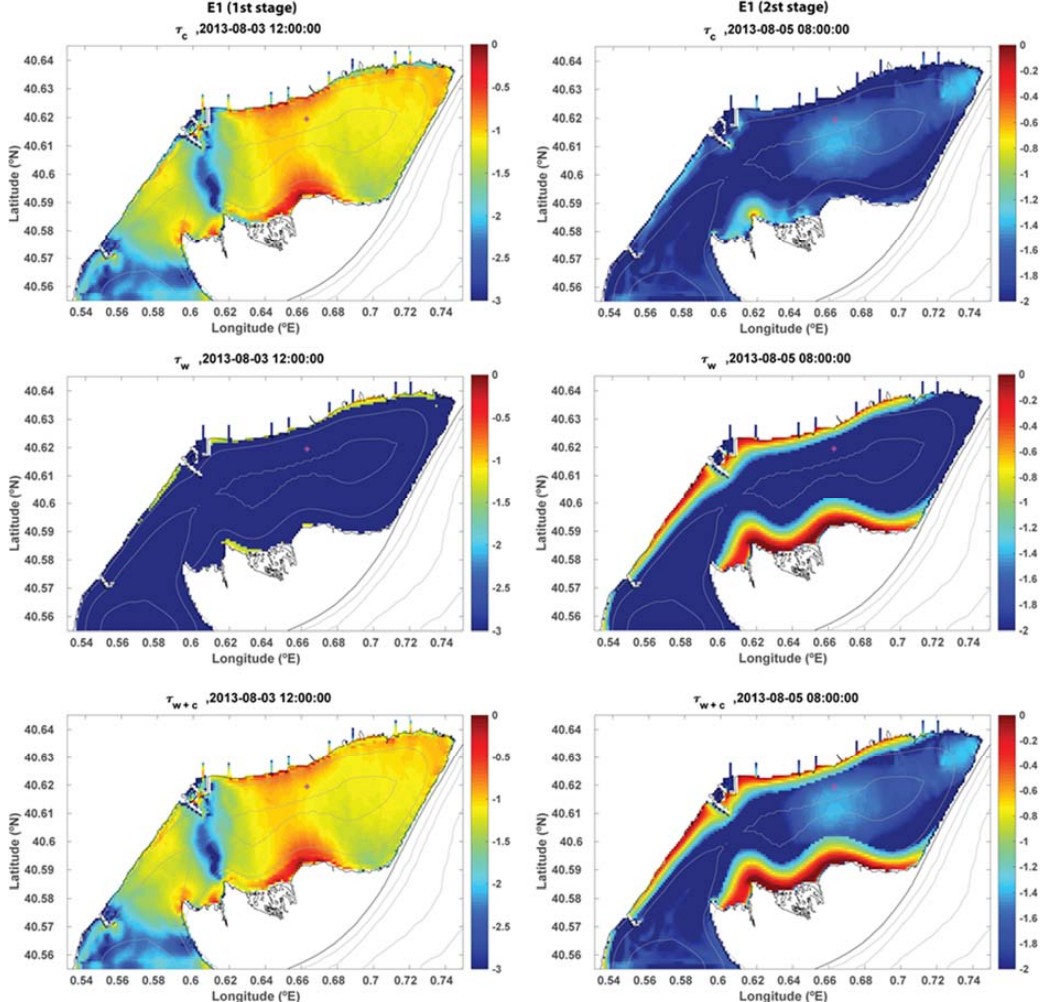

**Figure 5: Distribution of the current, wave and combined wave-current bottom stresses log10(Pa) in the Alfacs Bay during the first stage of the episode E1 (i.e. seiche) and the second stage of the episode E1 (i.e. sea breeze). Magenta dot show the A2 station. Isobaths (in grey) are plotted each 3 m. Note that for clarity, the plot scale is transformed in log10 and the vertical range differs between both bottom stress distributions.**




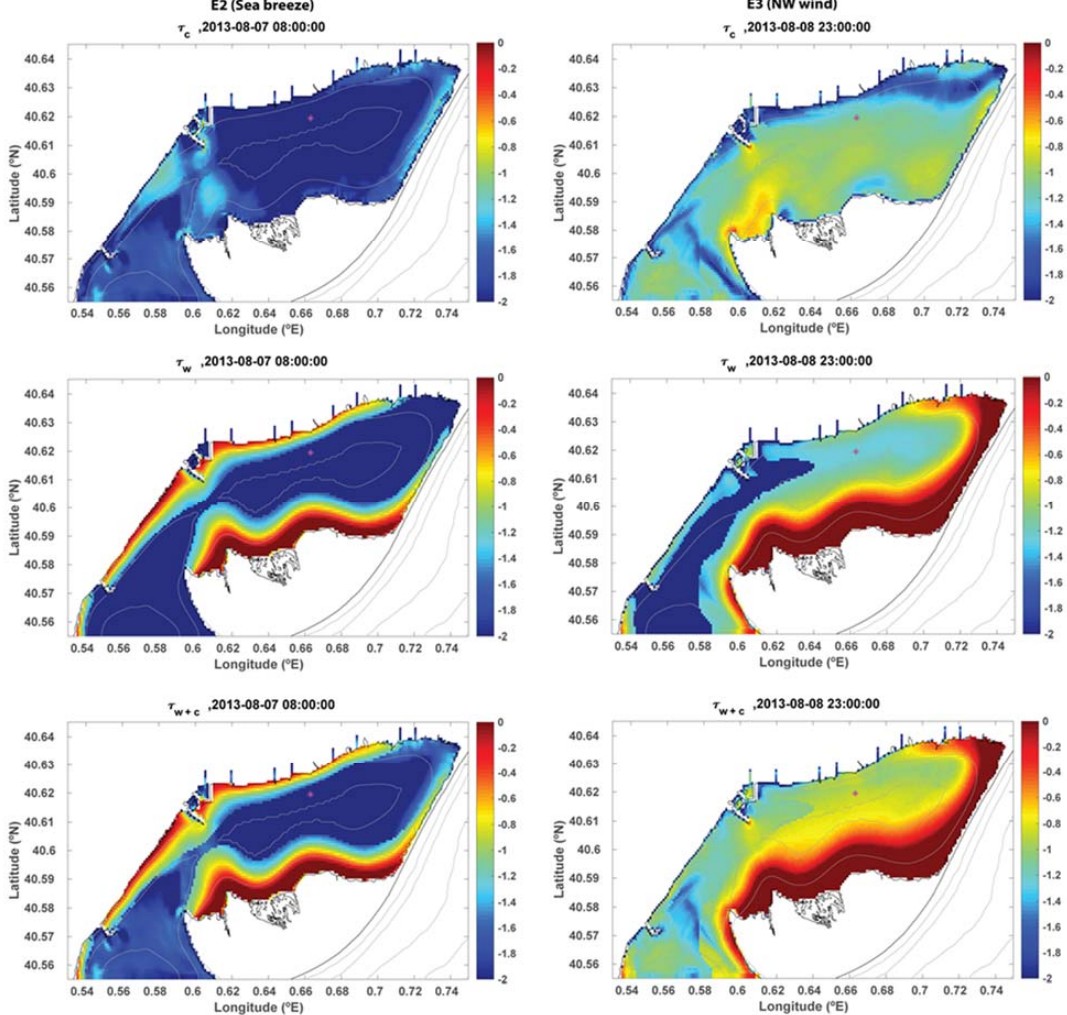

**Figure 6: Distribution of the current, wave and combined wave-current bottom stresses log10(Pa) in the Alfacs Bay during the first stage of the episode E2 and E3. Magenta dot show the A2 station. Isobaths (in grey) are plotted each 3 m. Note that for clarity, the plot scale is transformed in log10 and the vertical range differs between both bottom stress distributions.Figure 7: Each panel shows on the top the wind measured at M-Sc (in m·s⁻¹), and on the bottom the vertical profiles velocities measured at A2 (in m·s⁻¹). In each panel, different events are showed: 3/8/2013 for alongshore velocities (panel a) and 8/8/2013 for crosshore (panel b). Black lines show 0 velocity isolines.**






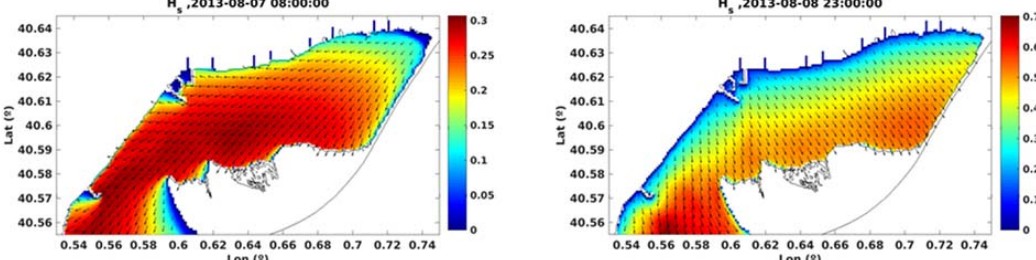

**Figure 7: Snapshot of the wave field for the episode E2 (sea-breeze) and E3 (NW wind). Color map represents the significant wave height and black arrows the direction of propagation. Note that the ranges of the significant wave height are different.**

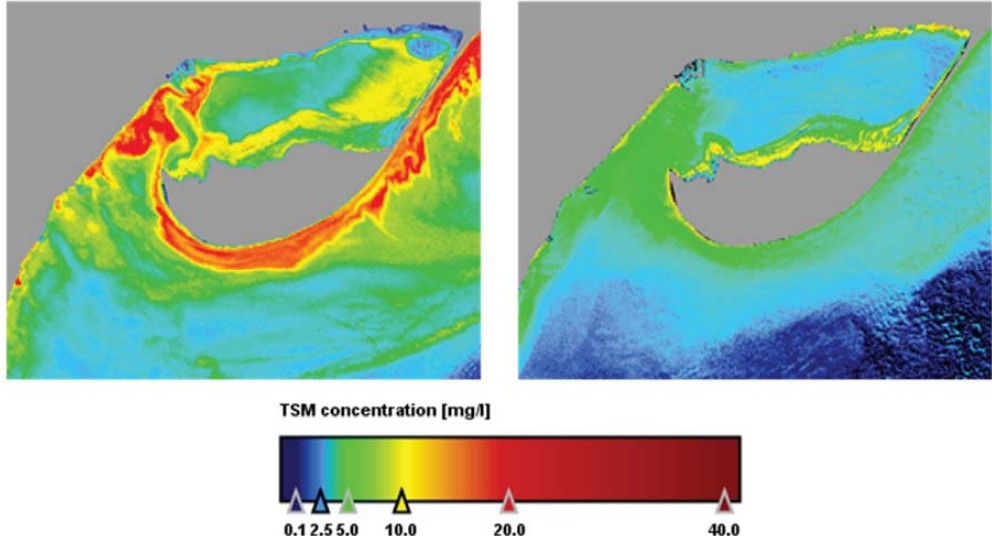

**Figure 8: Total Suspended Matter (TSM in mgr•l-1) obtained from Sentinel-2 for the Alfacs Bay in two differentiate scenarios: NW winds (left; 27th of December 2017) and calm conditions (right; 15th of February of 2018).**