# Peer review of "Characterization of bottom sediment resuspension events observed in a micro-tidal bay"

_Ocean Science, 2018_

## Referee Comment (RC1) · Anonymous Referee #1 · 9 Dec 2018

Manel Grifoll et al. presented hydrodynamic and turbidity data from both field observation and computational model simulation at Alfacs Bay (NW Mediterranean Sea) to investigate the sediment transport or resuspension mechanism in that area. The topic fits well with the themes of Ocean Science. The study would be a valuable knowledge input for future coastal management in that area. However, data presentations and interpretations, terminology and languages need to be improved a lot. Revision is suggested at this stage. Point-by-point review and comments are listed below. Some typos and sentences are listed, but are not limited to.

2. Methods 2.1 Study Area Line 75: Not sure is a typo or not. "shouthern (southern) spit" 2.2 Measurements campaigns It may be more straightforward if the authors use a table to show most of the sensor information. Line 106: How high is the wind monitor

above ground? 2.3 Current and wave model implementation Line 130: (10' data) not sure what it means. Or it's typo. Line 139: Equation (1), all terms in the equation need to be noted, e.g. u, v, etc. 3. Results 3.1 observations

Line 156-157: Terminology. "current speed" and "wind speed" are normally used in cientific papers, instead of "current intensity" and "wind intensity".

Line 161: Data for 1st and 6th October are not shown in the MS. If the authors prefer to make a statement about October observation, it would be better to add a supplemental figure for October observation. Otherwise, there is no support for the statement that "this seems evident during the 1st and 6th October".

Line 172-173: the data covering for August 3rd during E1 has been reported in the format of day and hour, but for events E2 and E3, the authors only report in the format of day without hour. In the figure, E2 and E3 do not cover the whole day on 6th August and 7th and 8th in August, respectively. I would suggest to report all three events in the format of day and hour.

Line 167: What is the reference for the sea level height, e.g. "sea level height in A1"? In figure 2(c), sea level height of 0m is referred to what?

(1) For figure2, if the authors add minor tick (in hour unit) for all the x-axis, which would be easier to follow when read the related interpretations. (2) For Figure2(b), y-axis could use major tick at 0, 90, 270, and 360, and minor ticks could set with an interval of 30, which would be easier to follow with the related interpretations. (3) Typos on the date expression, e.g. 3hd August, 3th August, which should be 3rd August.

3.2 Skill assessment near the sea bottom

Related Figure should be referred in this section (It should be Figure 4).

Line 179-180: Grammar mistake, "because they have ... and as a mechanism of re-suspension".

[Figure]

3.3 Modelled bottom stress

(1) Day and hour format are suggested to use for all events discussed. (2) Figures 5 and 6 have left panel and right panel, which are either different stages or different events, but in the data interpretation text, the authors refer to the entire figure, e.g. line 192 bottom stress (figure 5), in which the authors mean left panel of figure 5. The interpretation for figure 6 also has the same problem. It would be more readable if the authors make it clear which panel they are discussing. (3) Typos for the date in both text and figure 5 title.

Line 195-196: grammar mistake for this sentence.

Line 204: Figure 7 is cited here for E1, without any statement why E2 condition is used to explain E1 before the figure 7 is cited.

4. discussion (1) Line224-225: Clarify which panel of figure 5 is discussed here. (2) Line 251: typo. "corresponds". (3) Line 254-255: Clarify which panel in Figure 6 is discussed here. In addition, this sentence is difficult to follow. (4) Would suggest the authors to rewrite sentences at Line 235-236, 245-247,254-255, 273-275, 286-287, 318-319.

---

## Referee Comment (RC2) · Anonymous Referee #2 · 11 Dec 2018

The manuscript is focused on the causes of resuspension events at Alfacs Bay and compare the results of the model with near-botom turbidity observations . The application of the model may be useful for the area of study but various revisions need to be done to improve the quality of grammar and figures. It is hard to read the paper in its current state.

Specific comments: Line 28: Replace "convey" by "transports" Line 34: Finally, "the growth of" harmful species Line 38: In coastal areas, remove "of fate of" ; replace "pattern" by conditions. Line 39: "In" with "as a function" of ; Is "Sedimentological" a term that is used ? Line 40 "Such as" by "driven by" wind -waves Line 44 "reworking" by transports" Line 46 "at the water", by "in the water" Line 53: This is a suggestion to move "Thus, Alfacs Bay...." line after the end of Line 49. Line 60 " Driven" by driving

Line 69" Has" by Have Line 69: Improve this entire line after "Synchronous optical .." for better understanding. Line 83: is composed "of" Line 83: Use technical terms "sandy mud" ?

Linr 97: corresponds to the two months of and remove "correspond to summer conidtions" Figure 1 " Remove the background grid in Figure d, label the colorbar Line 122: Citations should be in a chronological order. Repetition of text in section 2.3. Carefully edit. Replace regular grid with curvilinear grid. Line 128: What are the time steps in two models. Line 132 10m3 → correction Line 136 as a fucnction Line 147: What do you mean by eddy profiles are scaled in wave boundary layer following and iterative processes? No need to write the equations if they are being not used in explaining the results. The citations are sufficient.

Line 157 What is M-Sc. Line 161 : October ? Line 164 : 3rd of August Line 233: Replace differentiate with different. Missing labels on Figure 2 and caption Figure 5: It is hard to see the magenta dot, use a symbol for locations.

---

## Author Comment (AC1) · 15 Jan 2019

Manel Grifoll et al. presented hydrodynamic and turbidity data from both field observation and computational model simulation at Alfacs Bay (NW Mediterranean Sea) to investigate the sediment transport or resuspension mechanism in that area. The topic fits well with the themes of Ocean Science. The study would be a valuable knowledge input for future coastal management in that area. However, data presentations and interpretations, terminology and languages need to be improved a lot. Revision is suggested at this stage. Point-by-point review and comments are listed below. Some typos and sentences are listed, but are not limited to.

The authors acknowledge the helpful comments and corrections of Referee #1, which helped to improve the quality of the manuscript. Below, each comment is answered point-by-point. The new version of the manuscript is also included.

2. Methods 2.1 Study Area Line 75: Not sure is a typo or not. "shouthern (southern) spit"

Corrected. Thanks

2.2 Measurements campaigns It may be more straightforward if the authors use a table to show most of the sensor information.

We built the table, but we think that the information is redundant with the text. We decide to avoid it.

| Station | Devices | Parameters |
|---------|-----------|-------------------------------|
| A1 | ADCP | Sea level height, water current |
| | OBS | NTU |
| A2 | ADCP | Sea level height, water current |
| | OBS | NTU |
| M-Sc | Anemometer | Wind speed and direction |

Line 106: How high is the wind monitor above ground?

m above the ground. This information is included in the new version of the Manuscript.

2.3 Current and wave model implementation

Line 130: (10' data) not sure what it means. Or it's typo.

Replaced by 600 sec (S.I)

Line 139: Equation (1), all terms in the equation need to be noted, e.g. u, v, etc.

Done

3. Results

3.1 observations

Line 156-157: Terminology. "current speed" and "wind speed" are normally used in scientific papers, instead of "current intensity" and "wind intensity".

Replaced in the new version of the manuscript and Figure 2.

Line 161: Data for 1st and 6th October are not shown in the MS. If the authors prefer to make a statement about October observation, it would be better to add a supplemental figure for October observation. Otherwise, there is no support for the statement that "this seems evident during the 1st and 6th October".

This was a mistake. Replaced "October" by "August".

Line 172-173: the data covering for August 3rd during E1 has been reported in the format of day and hour, but for events E2 and E3, the authors only report in the format of day without hour. In the figure, E2 and E3 do not cover the whole day on 6th August and 7th and 8th in August, respectively. I would suggest to report all three events in the format of day and hour.

Ok, the three events are described in the format day and hour.

Line 167: What is the reference for the sea level height, e.g. "sea level height in A1"?

A clarification is introduced in the new version of the manuscript: "*The sea level height reference was obtained subtracting the mean value of the pressure meter time-series provided by the ADCP.*".

In figure 2(c), sea level height of 0m is referred to what?

See previous comment.

(1) For figure2, if the authors add minor tick (in hour unit) for all the x-axis, which would be easier to follow when read the related interpretations.

After several editing proofs 6 hours minor tick is included in the Figures.

(2) For Figure2(b), y-axis could use major tick at 0, 90, 270, and 360, and minor ticks could set with an interval of 30, which would be easier to follow with the related interpretations.

Modified. Interval of 45º is included as a minor tick.

(3) Typos on the date expression, e.g. 3hd August, 3th August, which should be 3rd August.

All of them corrected. Thanks.

3.2 Skill assessment near the sea bottom

Related Figure should be referred in this section (It should be Figure 4).

Included. Thanks.

Line 179-180: Grammar mistake, "because they have . . . and as a mechanism of resuspension".

Corrected.

3.3 Modelled bottom stress (1) Day and hour format are suggested to use for all events discussed.

Ok, included in the new version of the manuscript.

(2) Figures 5 and 6 have left panel and right panel, which are either different stages or different events, but in the data interpretation text, the authors refer to the entire figure, e.g. line 192 bottom stress (figure 5), in which the authors mean left panel of figure 5. The interpretation for figure 6 also has the same problem. It would be more readable if the authors make it clear which panel they are discussing.

The specification of the panel "left" and "right" is included in the new version of the manuscript. Also the Figure's caption 5 and 6 include the "left" and "right" to be consistent with the text.

(3) Typos for the date in both text and figure 5 title.

Corrected in both cases

Line 195-196: grammar mistake for this sentence.

Corrected.

Line 204: Figure 7 is cited here for E1, without any statement why E2 condition is used to explain E1 before the figure 7 is cited.

Figure 7 is cited because represents the wave field during sea breeze condition, which occurs during the second stage of E1.

4. discussion (1) Line224-225: Clarify which panel of figure 5 is discussed here.

Clarified.

(2) Line 251: typo. "corresponds".

Corrected. Thanks.

(3) Line 254-255: Clarify which panel in Figure 6 is discussed here. In addition, this sentence is difficult to follow.

Clarified. The sentences have been modified: "According to the bottom stress shown in Figure 6(right) the wave induced bottom stress prevails. However, the complexity of the resuspension mechanisms, which the advection may have a relevant role, difficult to quantify the relative importance of each resuspension mechanism (i.e. wind or waves)."

(4) Would suggest the authors to rewrite sentences at Line 235-236, 245-247,254-255, 273-275, 286-287, 318-319.

All of these sentences have been rewritten to clarify.

[revised manuscript text omitted]

---

## Author Comment (AC2) · 15 Jan 2019

The manuscript is focused on the causes of resuspension events at Alfacs Bay and compare the results of the model with near-bottom turbidity observations. The application of the model may be useful for the area of study but various revisions need to be done to improve the quality of grammar and figures. It is hard to read the paper in its current state.

The authors acknowledge the helpful comments and corrections of Referee #2, which helped to improve the quality of the manuscript. Below, each comment is answered point-by-point. The new version of the manuscript is also included.

Specific comments:

Line 28: Replace "convey" by "transports" Line 34: Finally, "the growth of" harmful species Line 38: In coastal areas, remove "of fate of" ; replace "pattern" by conditions. Line 39: "In" with "as a function" of ;

All these comments have been included in the new version of the manuscript.

Is "Sedimentological" a term that is used ?

Replaced by "sediment" in the new version of the manuscript.

Line 40 "Such as" by "driven by" wind –waves

Corrected.

Line 44 "reworking" by transports"

We prefer to keep "reworking" because is used by seminal papers in re-suspension process (e.g. Wright and Nittrouer, 1995).

Line 46 "at the water", by "in the water" Line 53:

Done.

This is a suggestion to move "Thus, Alfacs Bay...." line after the end of Line 49.

Suggestion accepted.

Line 60 " Driven" by driving Line 69" Has" by Have

Both comments accepted.

Line 69: Improve this entire line after "Synchronous optical .." for better understanding.

Replaced "chance to advance in the interpretation of resuspension" by "chance to investigate the resuspension"

Line 83: is composed "of"

Ok, done.

Line 83: Use technical terms "sandy mud" ?

The sentence have been modified: "Alfacs is composed of mud, with significant content of clay, and sand (Palacín et al., 1991)"

Linr 97: corrsponds to the two months of and remove "correspond to summer conidtions"

Done.

Figure 1 " Remove the background grid in Figure d, label the colorbar

The background grid correspond to the numerical mesh used in the numerical modelling. As we think that this information is useful for scientist working in similar environments, we have included a clarification in the figure caption:

*"The background grid corresponds to the numerical mesh used in the numerical modelling."*

Line 122: Citations should be in a chronological order.

Done.

Repetition of text in section 2.3. Carefully edit.

We couldn't identify the text repetition in Section 2.3.

Replace regular grid with curvilinear grid.

Done

Line 128: What are the time steps in two models.

Information included in the new version of the manuscript:

*"The interval time between change of variables of ROMS and SWAN was established in 3600 s."*

Line 132 10m3 ! correction Line 136 as a fucnction

Both corrections included. Thanks.

Line 147: What do you mean by eddy profiles are scaled in wave boundary layer following and iterative processes? No need to write the equations if they are being not used in explaining the results. The citations are sufficient.

Ok, the sentence has been deleted to avoid confusion. The proper reference is provided (e.g. Kumar et al., 2012).

Line 157 What is M-Sc.

Location of the Meteorological Station (see Figure 1). The manuscript is clarified.

Line 161 : October ?

It was typo. Corrected. Thanks.

Line 164 : 3rd of August Line 233: Replace differentiate with different.

Both suggestions included.

Missing labels on Figure 2 and

Labels included in the new version of the figure.

caption Figure 5: It is hard to see the magenta dot, use a symbol for locations.

A symbol for A2 station included in the new version of the figure.

**References** (also included in the manuscript)

[revised manuscript text omitted]

---

## Referee Report (RR1)

The authors have replied point-by-point to all comments given by the referees. Most of them addressed the issues properly. However, as the two referees mentioned, quite lots of effort are needed to improve the grammar and the quality of this manuscript. This does not mean the rest of the text apart from pointed comments listed by the referees are clear and precise enough for readers. There are still a lot of grammar mistakes and incorrect way of forming sentences for a scientific publication. I have listed some of them as the following comments. Thus, I suggest a close check through the text and a major revision of the current version.

Line 24: at -> in a

Line 28: Also -> Futhermore

Line 32: sedimentation -> sediment

Line 37: with -> to

Line 39: remove 'redistribute

Line 40: I am wondering if there are any studies existed in this area focusing on sediment transport due to density gradient? Water circulation caused by density gradient usually plays an important role in transport of sediment and formation of turbidity maixima in estuaries.

Line 41, 42: list references in a chronological order. (Please check through the manuscript)

Line 46: remove the second "("

Line 54-55: Text duplicated, please remove.

Line 64-65: rewrite: "; the knowledge of these ⋯" -> ", which may further benefit human activities management ⋯"

Line 77: add "m" behind 6.5

Line 78: change "6.5m" to "6.5 m"

Line 108: put (M-Sc) behind land station.

Line 126: consisted -> consists

Line 128: in (Cerralbo et al., 2015a) -> by Cerralbo et al. (2015a)
(Check through the manuscript for similar isssues)

Line 130: Rewrite this sentence. (e.g., The barotropic time step for ROMS is set to 30 s, and in SWAN the wave field is solved in a time interval of 3600 s.)

Line 131: change -> exchange?

Line 131: open boundary was forced -> boundary coan't be forced. -> water motion at the open boundaries was forced by ⋯

Line 134: flow -> discharge

Line 135-136: This sentence is unclear, please rewrite.
Line 137: What are the velocity near the bottom and wave near the bottom?

Line 139: move "bottom stress" behind current and wave bothd.

Line 144: u and v are current spped in ? and ? direction? What is z_0 in your equation (1)? and the value is?

Line 153: (Kumar et al., 2012) -> Kumar et al. (2012)

Line 158: change "in" A2 to "at" A2. Use "at" for stations. Please close check you rest text and figure captions.

Line 159: measured sea level height?

Line 160: This sentence is still unclear, please rewrite it.

Line 162: change to "Two typical wind conditions are considered ⋯"

Line 168: remove "," and include -> "includes"

Line 169: The sentence is too wordy. Please remove "characteristic"

Line 171: it is unclear the amplitude of what is maximum?

Line 173: it is unclear by analyzing what in the along-shore direction of Figure 3 reveals the peak of velocities are at the order of 0.5 m s$^{-1}$. Note that the along-shore direction only refers to coordinate, itself can't reveal anything, please be precise in your description.

Line 176: ⋯ behavior ranging values from ⋯ -> ⋯ behavior with values ranging from ⋯

Line 176: I guess you mean Figure 2e.

Line 177: In this sence, three differentiated ⋯ -> Three differentiated ⋯

Line 183: works -> work

Line 188: please explain what are the "points" mean ?

Line 189: First, please explain what are shown in this figure. I don't see "points", but two triangles that represent current data analised from station $A_1$ and $A_2$, respectively, and one circle lying on the bottom of this figure reads "ADCP". Without explanation, it is very hard to understand what can you read from this figure.

Line 193: add "the" before Alfacs Bay. Remove "The" before Figures 5. Moreover, it is better to describe two figures separately. First Figure 5, then Figure 6.

Line 194: different snapshots of what? Please be precise on what you what to show.

Line 194: both -> two.

Line 194: wave and current-induced bottom ⋯ -> wave-induced and current-induced bottom⋯ And also please be consists through the entire text.

Line 195: corresponds -> correspond (Please rewrite this sentence in correct grammar.)

Line 196: removed "combined". It is clear that the bottom stress contains several components that due to different hydrodynamics, in which one or more dominant over the others.

Line 197: "stresses" -> "stress"; "due to the current bottom stress" -> "caused by currents"

Line 202: remove "to"

Line 205: add ")," behind E1, remove "where", "increase" -> "increases"

Line 206-208: Rewrite. This sentence is not an English written for scientific publication.

Line 209: "This figure shows how⋯" -> "It reveals that ⋯"

Line 228: not the modelled stress itself suggest ⋯ but the analysis of modelled stress. This is a problem frequently occurs in the manuscript. Please look into your grammar closely and fix it.

Line 231-232: In general, your sentences are wordy. Here is just one example. Moreover, it does not follow the structure formed in previous sentence. I suggest to rewrite as follows: "However, these studies did not explain the high spatial variability of the seiche-induced sediment resuspension, which are implied by the modelled current-induced bottom stress."

Line 229-232: Please rewrite and improve your way of writing sentences.

Line 238: "the current bottom intensity measured" -> "the measured bottom current speed"

Line 238: (Llebot et al., 2014) and (Cerralbo et al. 2015a) -> Llebot et al. (2014) and Cerralbo et al. (2015a)

Line 239: ··· a barotropic shape of what?

Line 240: behavior -> Please be consists with your spelling, use either British English or American English. Don't use them both in one manuscript. I observed "analyzed", "modeled", which are AE, while "behaviour", "modelled" are BE.

Line 245: "This" -> "The"

Line 246-248: Rewrite. Inconsistent structure.

Line 249-251: wordy sentence.

Line 256: similar to what in the second stage of E1.

Line 256: are -> is

Line 256: This sentence is unclear. Please rewrite.

Line 259: intensity -> speed

Line 261: have a relevant role in the resuspension mechanisms (wordy) -> relevant to the resuspension.

Line 262: remove "the"

Line 266: "the relative importance" with respect to what? To each other or to the combined bottom stress?

Line 267: quantify -> be quantified.

Line 271-272: I don't see how model data is correlated with filed data. Model is, to some extent, to mimic features you observed in field data with giving open boundaries and initial conditions.

Line 275: mechanism itself does not have spatial and temporal variability, but the relative importance of each mechanism does. Mechanisms refer to, not physical variables, but processes as explanations of a phenomenon.

Line 279: "an evident influence" -> what is that?

Line 284-286: Grammar incorrect, thus hard to understand.

Line 292-293: don't use the same word through the entire text. Moreover, two "contribution" have different meaning. I suggest to replace the first one with "study"

Line 296: add "those considering" behind including.

Line 297: "must take into account" -> "should include"
Line 300: The bay geometry characteristics cannot suggest. Please rewrite this sentence.

Line 300: remove "effect"

Line 304: remove "the"

Line 305: "This may be consistent with …" Please rewrite this sentence.

Line 308: "should allow" -> "allows"

Line 320: "Others" -> "Other"

Line 321: add "with" before "freshwater"

Figure 2 caption: "intensity" -> "speed"; "in"-> "at";

Line 509: "velocity" -> "speed". Note velocity refers to both speed and direction.
Line 510: "in" -> "by"

Figure 3 caption: (a)…; (b) as (a), but for the cross-shore direction.

"showed" -> "shown"

Figure 5 caption: current-induced bottom stress (\tau_c), wave-induced bottom stress (\tau_w) and combined wave-current bottom stress (\tau_c + \tau_w).

Magenta is not very clear in the contour plots. I suggest to use a color with better contrast.

"Isobaths (in grey) are plotted each 3 m" -> Isobaths are plotted in grey solid lines in 3 m interval from ? m to ? m.

"the plot scale is transformed in log10" -> "the bottom stresses are plotted in log10 scale"

---

## Author Response (AR2)

The authors have replied point-by-point to all comments given by the referees. Most of them addressed the issues properly. However, as the two referees mentioned, quite lots of effort are needed to improve the grammar and the quality of this manuscript. This does not mean the rest of the text apart from pointed comments listed by the referees are clear and precise enough for readers. There are still a lot of grammar mistakes and incorrect way of forming sentences for a scientific publication. I have listed some of them as the following comments. Thus, I suggest a close check through the text and a major revision of the current version.

Thank you very much for your comments. We have improved the English writing in the revised manuscript. I think that all spelling and grammatical errors in the manuscript have been addressed. These changes will not influence the content of the paper. Therefore here, we did not list the changes, but those revisions within the document are presented as "tracked changes".

Line 24: at -> in a

Done

Line 28: Also -> Futhermore

Done

Line 32: sedimentation -> sediment

Corrected

Line 37: with -> to

Done

Line 39: remove 'redistribute

Done

Line 40: I am wondering if there are any studies existed in this area focusing on sediment transport due to density gradient? Water circulation caused by density gradient usually plays an important role in transport of sediment and formation of turbidity maixima in estuaries.

The water circulation caused by density gradients is expected to be small and presumably not able to resuspend fine bottom sediment which is the topic of the article. There is many references addressing the gravitational circulation (e.g. Gyer, 2014 in Continental Shelf Research) is a good review), but we avoid to mention to not confuse the reader.

Line 41, 42: list references in a chronological order. (Please check through the manuscript)

Corrected through the manuscript.

Line 46: remove the second "("

Done

Line 54-55: Text duplicated, please remove.

Removed

Line 64-65: rewrite: "; the knowledge of these …" -> ", which may further benefit human activities management …"

modified

Line 77: add "m" behind 6.5

Done

Line 78: change "6.5m" to "6.5 m"

Changed.

Line 108: put (M-Sc) behind land station.

Done

Line 126: consisted -> consists

Corrected

Line 128: in (Cerralbo et al., 2015a) -> by Cerralbo et al. (2015a)

(Check through the manuscript for similar isssues)

Corrected through the manuscript.

Line 130: Rewrite this sentence. (e.g., The barotropic time step for ROMS is set to 30 s, and in SWAN the wave field is solved in a time interval of 3600 s.)

Rewrite in the new version of the manuscript.

Line 131: change -> exchange?

Corrected

Line 131: open boundary was forced -> boundary coan't be forced. -> water motion at the open boundaries was forced by …

Modified.

Line 134: flow -> discharge

Modified.

Line 135-136: This sentence is unclear, please rewrite.

Modified.

Line 137: What are the velocity near the bottom and wave near the bottom?

Near the bottom is the nearest computational node near the bottom. We include the reference where the computational method is explained to avoid extend unnecessarily the manuscript.

Line 139: move "bottom stress" behind current and wave bothd.

Done

Line 144: u and v are current spped in ? and ? direction? What is z_0 in your equation (1)? and the value is?

Both of them corrected. Z0 is computed by the model.

Line 153: (Kumar et al., 2012) -> Kumar et al. (2012)

Done.

Line 158: change "in" A2 to "at" A2. Use "at" for stations. Please close check you rest text and figure captions.

Done. Thanks.

Line 159: measured sea level height?

Surface elevation, we think that next sentence is explained.

Line 160: This sentence is still unclear, please rewrite it.

modified

Line 162: change to "Two typical wind conditions are considered …"

Modified.

Line 168: remove "," and include -> "includes"

Done

Line 169: The sentence is too wordy. Please remove "characteristic"

Modified and "Characteristic" removed.

Line 171: it is unclear the amplitude of what is maximum?

Clarified: "amplitude of sea level oscillation".

Line 173: it is unclear by analyzing what in the along-shore direction of Figure 3 reveals the peak of velocities are at the order of 0.5 m s-1. Note that the along-shore direction only refers to coordinate, itself can't reveal anything, please be precise in your description.

The along-shore is the axis direction of the Bay. See L171 in the new version of the Bay.

Line 176: … behavior ranging values from … -> … behavior with values ranging from …

Modified.

Line 176: I guess you mean Figure 2e.

Done.

Line 177: In this sence, three differentiated … -> Three differentiated …

Corrected.

Line 183: works -> work

Clarified.

Line 188: please explain what are the "points" mean ?

Clarified.

Line 189: First, please explain what are shown in this figure. I don't see "points", but two triangles that represent current data analised from station A1 and A2 , respectively, and one circle lying on the bottom of this figure reads "ADCP". Without explanation, it is very hard to understand what can you read from this figure.

Clarified. The sentence has been expanded.

Line 193: add "the" before Alfacs Bay. Remove "The" before Figures 5. Moreover, it is better to describe two figures separately. First Figure 5, then Figure 6.

Done. In the manuscript the figures are explained sequentially.

Line 194: different snapshots of what? Please be precise on what you what to show.

Modified by results.

Line 194: both -> two.

Modified.

Line 194: wave and current-induced bottom … -> wave-induced and current-induced bottom… And also please be consists through the entire text.

Corrected through the manuscript

Line 195: corresponds -> correspond (Please rewrite this sentence in correct grammar.)

Corrected.

Line 196: removed "combined". It is clear that the bottom stress contains several components that due to different hydrodynamics, in which one or more dominant over the others.

Done.

Line 197: "stresses" -> "stress"; "due to the current bottom stress" -> "caused by currents"

Corrected,

Line 202: remove "to"

Corrected,

Line 205: add ")," behind E1, remove "where", "increase" -> "increases"

Done,

Line 206-208: Rewrite. This sentence is not an English written for scientific publication.

Modified: *"At A2, the combined bottom stress is equal to 0.03 Pa (presumably too small to induce resuspension)."*

Line 209: "This figure shows how…" -> "It reveals that …"

Modified.

Line 228: not the modelled stress itself suggest … but the analysis of modelled stress. This is a problem frequently occurs in the manuscript. Please look into your grammar closely and fix it.

Corrected and checked.

Line 231-232: In general, your sentences are wordy. Here is just one example. Moreover, it does not follow the structure formed in previous sentence. I suggest to rewrite as follows: "However, these studies did not explain the high spatial variability of the seiche-induced sediment resuspension, which are implied by the modelled current-induced bottom stress."

Modified.

Line 229-232: Please rewrite and improve your way of writing sentences.

Ok, done.

Line 238: "the current bottom intensity measured" -> "the measured bottom current speed"

Corrected.

Line 238: (Llebot et al., 2014) and (Cerralbo et al. 2015a) -> Llebot et al. (2014) and Cerralbo et al. (2015a)

Done.

Line 239: … a barotropic shape of what?

Added: *"water current profiles"*

Line 240: behavior -> Please be consists with your spelling, use either British English or American English. Don't use them both in one manuscript. I observed "analyzed", "modeled", which are AE, while "behaviour", "modelled" are BE.

Ok, checked through the manuscript.

Line 245: "This" -> "The"

Corrected

Line 246-248: Rewrite. Inconsistent structure.

Modified: "Episode E2 is attributed to at sea-breeze mechanism."

Line 249-251: wordy sentence.

Modified,

Line 256: similar to what in the second stage of E1.

modified

Line 256: are -> is

Modified.

Line 256: This sentence is unclear. Please rewrite.

Corrected,

Line 259: intensity -> speed

Corrected,

Line 261: have a relevant role in the resuspension mechanisms (wordy) -> relevant to the resuspension.

Modified.

Line 262: remove "the"

Line 266: "the relative importance" with respect to what? To each other or to the combined bottom stress?

Clarified

Line 267: quantify -> be quantified.

Modified,

Line 271-272: I don't see how model data is correlated with filed data. Model is, to some extent, to mimic features you observed in field data with giving open boundaries and initial conditions.

We have difficulties to understand this point, here we not compare model and observations. However, the sentence has been modified in the new version of the manuscript.

Line 275: mechanism itself does not have spatial and temporal variability, but the relative importance of each mechanism does. Mechanisms refer to, not physical variables, but processes as explanations of a phenomenon.

Thanks for this clarification. The manuscript has been modified.

Line 279: "an evident influence" -> what is that?

Modified by "*apparent*"

Line 284-286: Grammar incorrect, thus hard to understand.

Corrected and modified.

Line 292-293: don't use the same word through the entire text. Moreover, two "contribution" have different meaning. I suggest to replace the first one with "study"

Done.

Line 296: add "those considering" behind including.

Done

Line 297: "must take into account" -> "should include"

Replaced

Line 300: The bay geometry characteristics cannot suggest. Please rewrite this sentence.

Sentence modified: "favor" instead of "suggest".

Line 300: remove "effect"

Done.

Line 304: remove "the"

Done.

Line 305: "This may be consistent with …" Please rewrite this sentence.

Modified: *"This could explain…"*

Line 308: "should allow" -> "allows"

Replaced.

Line 320: "Others" -> "Other"

Corrected,

Line 321: add "with" before "freshwater"

Added

Figure 2 caption: "intensity" -> "speed"; "in"-> "at";

Done,

Line 509: "velocity" -> "speed". Note velocity refers to both speed and direction.

Corrected,

Line 510: "in" -> "by"

Corrected.

Figure 3 caption: (a)…; (b) as (a), but for the cross-shore direction.

Modified.

"showed" -> "shown"

Corrected

Figure 5 caption: current-induced bottom stress (\tau_c), wave-induced bottom stress (\tau_w)

and combined wave-current bottom stress (\tau_c + \tau_w).

Corrected.

Magenta is not very clear in the contour plots. I suggest to use a color with better contrast.

After a reviewer suggestion we include magenta because this color is not included in the color bar.

[revised manuscript text omitted]

---

## Author Response (AR3)

The authors acknowledge the helpful comments suggested by the Referee, which helped to improve the quality of the manuscript. Below, each comment is answered point-by-point.

1. In Abstract (line 22), the authors suggest that 'the previous sediment resuspension events may influence the increase of suspended sediment in later on events'. This is absolutely correct but does not make any sense. Please state precisely and consolidate whether the influence on the increase of SSC is positive or negative. In other words, will the previous resuspension enhance or suppress the increase of the subsequent resuspension?

Corrected: "*We suggest that the sequence of resuspension events plays a relevant role in suspended sediment concentration, in such a way that the occurrence of sediment resuspension events may increase the suspended sediment in subsequent events*"

2. Line 133, ... are described in detail in what?

Corrected

3. Line 142: please explain what do 'x' and 'y' indicate? Longitudinal and lateral?

Modified: "(*in both eastward and northward grid directions*)"

4. Line 172: add "," after 'Bay'

Included.

5. Section 'Conclusions' is quite loose. It reads more like a part of the Abstract. Please indicate clearly how do the seiching activity and wind-wave interaction affect the variability of the bottom sediment resuspension and the distributions of near-bottom turbidity? How does the previous resuspension affect the subsequent resuspension events? Be more consolidate in the conclusion.

Thanks, the conclusions has been extended and clarified in the new version of the manuscript.